# Distinct Elevational Patterns and Their Linkages of Soil Bacteria and Plant Community in An Alpine Meadow of the Qinghai–Tibetan Plateau

**DOI:** 10.3390/microorganisms10051049

**Published:** 2022-05-19

**Authors:** Jing Cong, Wei Cong, Hui Lu, Yuguang Zhang

**Affiliations:** 1College of Marine Science and Biological Engineering, Qingdao University of Science Technology, Qingdao 266042, China; yqdh77@163.com; 2Institute of Forest Ecology, Environment and Protection, Chinese Academy of Forestry and the Key Laboratory of Biological Conservation of State Forestry Administration, Beijing 100091, China; cong0915wei@163.com (W.C.); susanluhui@163.com (H.L.)

**Keywords:** soil microbiota, plant community, elevation, climate warming, linkage, GeoChip 4.0, 16S rRNA sequencing

## Abstract

Soil microbes play important roles in determining plant community composition and terrestrial ecosystem functions, as well as the direction and extent of terrestrial ecosystem feedback to environmental changes. Understanding the distribution patterns of plant and soil microbiota along elevation gradients is necessary to shed light on important ecosystem functions. In this study, soil bacteria along an elevation gradient in an alpine meadow ecosystem of the Qinghai–Tibetan Plateau were investigated using Illumina sequencing and GeoChip technologies. The community structure of the soil bacteria and plants presented a continuous trend along the elevation gradient, and their alpha diversity displayed different distribution patterns; however, there were no linkages between them. Beta diversity of the soil bacteria and plants was significantly influenced by elevational distance changes (*p* < 0.05). Functional gene categories involved in nitrogen and phosphorus cycling had faster changes than those involved in carbon degradation, and functional genes involved in labile carbon degradation also had faster variations than those involved in recalcitrant carbon degradation with elevational changes. According to Pearson’s correlation, partial Mantel test analysis, and canonical correspondence analysis, soil pH and mean annual precipitation were important environmental variables in influencing soil bacterial diversity. Soil bacterial diversity and plant diversity had different distribution patterns along the elevation gradient.

## 1. Introduction

Soil microbe–plant linkages play key roles in regulating the direction and extent of terrestrial ecosystem responses to environmental change, as well as the composition of plant communities and terrestrial ecosystem functions [1,2,3,4]. Plant diversity is widely used to predict soil microbial diversity, and a more diverse plant community is associated with a more diversified soil microbial community [2,5]. Evidence is increasingly suggesting that soil microbes can influence plant diversity and community composition, because soil microbes are important in biogeochemical cycles and many ecosystem processes [6,7,8]. Free-living soil microbes might indirectly promote plant diversity by increasing the diversity of the available nutrient pools [9]. Both arbuscular mycorrhizal fungi and N-fixing rhizobia bacteria symbionts complement each other, resulting in increased plant diversity [10]. Therefore, the relationships between soil microbial diversity and plant diversity tend to be mutualistic. However, there is still inconsistency, especially for being somewhat disengaged from alpha to beta diversity [1,2,11,12]. As a result, our lack of knowledge of the underlying links between soil microbial diversity and plant diversity means that we still have much to learn [2].

Climate change is becoming one of the largest anthropogenic disturbances to human well-being and natural ecosystems, and it alters species distributions and inter-organism interactions [7,13]. Many researchers have paid attention to the effect of climate change on soil microbes, plants, and their linkages in terrestrial ecosystems, and their findings from experimental and observational studies have shown variable associations within single sites or across broad spatial scales [2,11,12,14,15,16]. However, the responses and interactions of soil microbes and plant communities to climate change, including warming and changing precipitation, remain unclear [1,17,18,19]. This could be due to the fact that: (i) most previous experiments have primarily focused on plant communities; (ii) there are technological limitations due to the high plasticity and complexity of microbial community in natural environments; (iii) although some controlled warming experiments have lasted over a decade, the majority of them are short term; (iv) numerous warming studies focused only on several temperature points without continuous gradient changes [19,20,21,22].

Elevation gradients can serve as analogs for climate change, characterized by biotic turnover over small geographic distances and dramatic changes in climate. They play a critical role in the progression of biogeographical and ecological studies, as well as the prediction of possible consequences of long-term climate change [2,15,23,24]. Therefore, elevation gradients may be used as powerful “natural experiments” to test theories on how ecosystems adapt to climate change and feedback [7,24,25,26].

The Qinghai–Tibetan Plateau is the largest and highest low-latitude plateau in the world [27], and it is considered as the world’s third “pole” [28]. An alpine meadow is widely distributed from the elevation of 3200–5300 m, covering more than 40% of the Qinghai–Tibetan Plateau area [29,30]. The alpine meadow is a large soil organic carbon (SOC) pool, accounting for 2.5% of global soil carbon pool with an SOC content (0–75 cm) of 23.2 Pg [31]. Therefore, the alpine meadow ecosystem is extremely sensitive to climate change, making it an ideal place for adopting a space-for-time substitution strategy to study the response of natural ecosystems to climate change along elevation, temperature, and precipitation gradients [32,33,34].

In this study, we analyzed soil microbial communities and explored different distribution patterns and their linkages with soil bacterial diversity and plants along an elevation gradient in an alpine meadow ecosystem of the Qinghai–Tibetan Plateau based on high-throughput Illumina sequencing and functional gene microarray (GeoChip). The aims were to determine: (1) how soil bacterial communities and plants are distributed along an elevation gradient in an alpine meadow ecosystem, (2) how plant and soil microbial diversity (alpha and beta diversity) are linked, and (3) how different environmental factors affect plant and soil bacterial diversity under climate change.

## 2. Materials and Methods

### 2.1. Study Sites and Sampling

The study sites were located in the Sanjiangyuan National Natural Reserve in Qinghai Province, China (97°40′22″–100°05′27″E, 34°08′16″–35°56′06″N), which lies in the heart of the Qinghai–Tibetan Plateau [35]. The annual mean air temperature and annual mean precipitation are 3.8–5.6 °C and 262.2–772.8 mm, respectively [36]. In July, 2012, six study sites were selected from alpine meadow systems with similar slopes and aspects from 3200 to 4800 m, with an elevation distance of about 200–300 m. The sites were named in accordance with their altitudes (SJY-3220, SJY-3490, SJY-3880, SJY-4140, SJY-4480, and SJY-4790), and their detailed site information is presented in Table 1.

A grid of 200 m × 200 m was set up, and a total of 13 sample plots with 1 m × 1 m were acquired at each site. We sampled with two transect belts along the horizontal and vertical axes, which were located at 5-, 10-, 20-, 50-, 100-, and 200-m distances from the original point. Soil samples were obtained at 0–10 cm soil depth in non-rhizosphere soil of each plot using the diagonal method [37], and ten soil cores (diameter 4 cm) were taken from each plot and mixed to yield about 400 g of soil. Stones and roots were removed from samples before completely mixing the samples from each plot. At each site, 13 soil samples were collected. About 100 g of soil was preserved at −80 °C for soil microbial community research, and the rest of soil was kept at room temperature for soil property measurements. These detailed analyses are described in Section 2.2, Section 2.4, Section 2.5 and Section 2.6. Plant diversity and properties, including plant number, plant species, plant height, and canopy of each grass, were investigated in each plot using stand species identification and survey methods.

### 2.2. Measurement of Soil Physicochemical Characteristics

The SOC, total nitrogen (TN), available nitrogen (AN), total phosphorus (TP), available phosphorus (AP), soil pH, and soil moisture (Mo) were measured as previous study [38]. SOC and TN were determined by the wet oxidation and modified Kjeldahl procedures, soil pH was measured based on a pH meter with the glass electrode at a water to soil ratio of 2.5:1, and TP was assessed with a wet digestion method with concentrated HF and HClO4 [38]. Plant and soil physicochemical properties are listed in Appendix A.

### 2.3. Climate Data

Climate data were collected by the WorldClim (www.worldclim.org, accessed on 14 February 2019) global climate dataset based on the recorded GPS location information in each sampling plot. Mean annual precipitation (MAP) and mean annual temperature (MAT) over 50 years (1950–2000) were used to represent climate warming at each site.

### 2.4. Soil Microbial DNA Extraction, Purification, and Quantification

According to the manufacturer’s instructions, soil microbial DNA was extracted by the Fast DNA Spin kit for soil (MP Biomedical, Carlsbad, CA, USA). The quality of purified DNA was analyzed at the ratios of absorbance at 260/230 nm and 260/280 nm with a Nanodrop ND-1000 Spectrophotometer on a 0.8% low-melting-point agarose gel (Nanodrop Technologies, Inc., Wilmington, DE, USA). Finally, microbial DNA was quantified by a FlUOstar Optima (BMG Labtech, Jena, Germany).

### 2.5. Illumina Sequencing of Bacterial 16S rRNA Gene Amplicons and Data Processing

The purified DNA as a template, as well as 5′- GTGCCAGCMGCCGCGGTAA-3′ (515F) and 5′- GGACTACHVGGGTWTCTAAT-3′ (806R) as the forward and reverse PCR primer sequences, were used to amplify the V4 hypervariable region of bacterial 16S rRNA gene. Reverse primers were coupled with a barcode sequence. The PCR amplification system was in a 25 μL reaction, containing 0.4 μM primers, 5 μL DNA, 2.5 μL AccuPrime PCR buffer II [39], and 0.1 μL Taq Polymerase. The detailed PCR amplification conditions followed those described by Ding et al. A Miseq platform (Illumina, San Diego, CA, USA) was used to perform 2 × 250 bp paired-end sequencing on PCR products [38,40].

The 16S rRNA gene sequencing data were demultiplexed based on their unique barcode sequence. Low-quality (average quality score < 20%) and ambiguous DNA reads (contain N) were trimmed and deleted. FLASH (version 1.0.0) with a phred-offset of 33 was used to integrate the forward and reverse reads into a whole sequence [41]. Chimeric sequences were detected by UCHIME (version 5.2.3) [42]. UCLUST was used to define the operational taxonomic units (OTUs) at a 97% similarity level [42]. Singletons were removed. Taxonomic identification of each phylotype was determined using the ribosomal database project classifier (version 2.12). The number of identified OTUs at different classification levels was counted. Random resampling per soil sample was processed using the lowest sequences number (10,000). Data processing was finished using the Galaxy sequencing pipeline at the Institute for Environmental Genomics, University of Oklahoma, USA (http://zhoulab5.rccc.ou.edu:8080/, accessed on 14 February 2019).

### 2.6. GeoChip Hybridization and Data Processing

GeoChip 4.0 was used to explore the potential key metabolic functions of soil microbes. GeoChip 4.0 contains about 82,000 oligonucleotide probes containing 141,995 microbial functional genes engaged in 410 gene categories involved in carbon (C), nitrogen (N), and phosphorus (P) cycling, as well as other biogeochemical processes. Detailed probe and function information for GeoChip is displayed on the website (http://ieg.ou.edu, accessed on 14 February 2019). Cy3 fluorescent dye was labeled on microbial DNA by a random priming approach [43]. DNA hybridization was carried out at 42 °C for 16 h in a hybridization station (MAUI, BioMicro Systems, Salt Lake City, UT, USA), and the arrays were scanned at full laser power and 100% photomultiplier tubes with a NimbleGen MS200 Microarray scan (Roche, Madison, WI, USA). Scanned images were processed by NimbleScan software [43].

All the raw GeoChip data were uploaded to the data analysis manager (GeoChip 4.0) (http://ieg.ou.edu/microarray/, accessed on 14 February 2019). GeoChip data were pre-processed as follows: (i) spots with a signal-to-noise ratio of less than 2.0 or signal intensity less than 1000 were deleted, (ii) functional genes that were found in less than 6 out of 13 replicate samples in each study site were removed, and (iii) functional gene signal intensity was normalized by dividing the average signal intensity value of each plot.

### 2.7. Statistical Analysis

Plant alpha diversity was calculated using Shannon index and Pielou index. Soil bacterial richness, Shannon index, and Simpson index were counted based on 16S rRNA sequencing data. Normalized signal intensity of functional gene category and gene family at the same level was summed. Beta diversity of soil bacteria and plants was calculated based on Bray–Curtis distance, and the linkage between them was analyzed. Principal coordinates analysis (PcoA) was used to assess plant and bacterial community structure. Partial Mantel tests and canonical correspondence analysis (CCA) were used to investigate the relationship between community structure and environmental variables. We used variation inflation factors (VIFs) to determine whether the variance in canonical coefficients was inflated by the presence of correlations with environmental factors while choosing an attribute for the CCA model. The VIFs were used to remove superfluous parameters step by step in the CCA modeling. The R software package (v.3.5.2) was used for all statistical analysis, except for Analysis of variance (ANOVA) and Pearson correlation by IBM SPSS statistic 19.0.

## 3. Results

### 3.1. Climate and Soil Physicochemical Characteristics

According to the 1950–2000 climate data from WorldClim, the MAT of the warmest quarter increased as the elevation decreased, from 4.9 °C at SJY-4790 to 11.6 °C at SJY-3220, while the MAP reduced from 417 mm at SJY-4790 to 319 mm at SJY-3220 (Appendix A). Therefore, both the MAT and MAP distinctly changed along the elevation gradient. In addition, the contents of SOC, TN, AN, and soil moisture showed similar trends to the MAP (Appendix A). Soil pH significantly increased as the elevation decreased, and it was positively correlated with the MAP. Therefore, soil physicochemical characteristics changed dramatically and showed discrepant patterns along the elevation gradient.

### 3.2. Distribution Pattern of Plant Diversity along Elevation Gradient

A total of 74 plant species were identified at six sites, belonging to 21 families and 51 genera. Cyperaceous plants, such as *Kobresia pygmaea*, *Kobresia schoenoides*, and *Kobresia humilis*, became the dominant species in high-and middle-elevation locations (SJY-4790, SJY-4480, SJY-4140, and SJY-3880), while Poaceae plants, such as *Poa annua*, *Stipa capillata*, and *Elymus nutans*, dominated in the low-elevation locations (SJY-3490 and SJY-3220) (Table 1). Plant Shannon index and Pielou index were calculated by the field survey data, and revealed no distribution pattern along the elevation gradient (Table 2). Of the six research locations, SJY-4480 and SJY-4140 had the highest plant Shannon indices, while SJY-3880 had the lowest Shannon and Pielou indices. PcoA analysis showed plant community structure differed between sites, and a continuous trend was developed across the axis1 from SJY-3220 to SJY-4790 (Figure 1a). Plant beta diversity was significantly affected by elevation distance (*r* = 0.696, *p* < 0.001, Appendix A). These findings revealed that plant alpha diversity had no elevation distribution patterns, while plant community structure and beta diversity were significantly changed along the elevation gradient in the alpine meadow ecosystem (*p* < 0.001).

### 3.3. Distribution Pattern of Soil Community Structure and Bacterial Diversity along Elevation Gradient

A total of 1,726,038 high quality sequences were obtained from grassland soil samples. These sequences were classified into 33,046 OTUs at 97% sequence identity threshold. At least 28 known bacterial phyla were detected by phylogenetic analysis. Soil bacterial community structure changed among different sites. *Proteobacteria* was the most dominant phylum from SJY-4790 to SJY-4140, accounting for over 29% of the relative abundance, while *Acidobacteria* was the most dominant phylum in SJY-3220 and SJY-3490, accounting for over 29% of the relative abundance (Figure 2). Along the elevation gradient, three different soil bacterial distribution patterns were observed: (i) the richness and relative abundance of *Proteobacteria*, *Verrucomicrobia*, and *Bacteroidetes* were considerably decreased as elevation decreased; (ii) the richness and relative abundance of *Acidobacteria* and *Actinobacteria* were dramatically increased as elevation decreased; (iii) some taxa, such as the *Gemmatimonadetes* and *Chlorobi*, were not changed along the elevation gradient (Appendix A). The relative abundance of dominant classes of soil bacteria (>0.1%) followed a similar pattern (Appendix A). For example, when elevation decreased, *Alpha*-*proteobacteria* and *Acidobacteria Gp4* were greatly increased, while *Gamma*-*proteobacteria* and *Acidobacteria Gp22* were decreased (Appendix A).

Soil bacterial alpha diversity was calculated by Shannon and Simpson indices, and OTU numbers (richness). Shannon index ranged from 7.26 ± 0.12 to 7.50 ± 0.08, Simpson index ranged from 448 ± 102 to 755 ± 145, and OTU richness ranged from 3144 ± 215 to 3517 ± 153 at these six sites (Table 2 and Appendix A). Soil bacterial alpha diversity was significantly higher (*p* < 0.05) at low-elevation sites (SJY-3220 and SJY-3490) than at high-elevation sites (SJY-4790, SJY-4480, SJY-4140, and SJY-3880). However, there were no differences among these high-elevation sites (SJY-4790, SJY-4480, SJY-4140, and SJY-3880) (Table 2 and Appendix A). PcoA showed that soil bacterial community structure presented a continuous trend across the axis1 from SJY-3220 to SJY-4790 (Figure 1b). Bacterial beta diversity was significantly affected by elevation distance changes (*r* = 0.865, *p* < 0.001, Appendix A). These results showed that soil bacterial community structure and diversity significantly altered along the elevation gradient in the alpine meadow ecosystem.

### 3.4. Soil Microbial Functional Genes Involved in C, N, and P Cycles

Soil microbial functional genes involved in C, N, and P cycling were analyzed by GeoChip 4.0. There were 23,898, 6698, and 2146 genes found in response to C, N, and P cycling, respectively. The relative abundance of functional gene categories varied among different soil sampling sites; however, no elevation trends were observed (Figure 3). Total relative abundance of functional gene categories involved in N cycling, C fixation, methane metabolism, and P cycling at SJY-3220 and SJY-3490 were significantly higher than at SJY-4790 (*p* < 0.1), while functional gene categories involved in SOC degradation remained changed (Figure 3).

Gene categories produced comparable results at the functional gene level. Most functional genes involved in soil N (*nifH*, *amoA*, *nosZ*, and *nirK*) and P (*ppk* and *ppx*) cycling increased in relative abundance as elevation decreased, and they were significantly higher at SJY-3220 and SJY-3490 than at the high-elevation sites (SJY-4790, *p* < 0.1, Appendix A). Many genes involved in labile C degradation (*starch*, *cellulose*, and *hemicellulose*) at SJY-4790 were significantly higher compared to the low-elevation sites. For example, *xylanase* gene involved in hemicellulose degradation was significantly higher at SJY-3490 (*p* = 0.092) and SJY-3220 (*p* = 0.028) than at SJY-4790 (Appendix A). However, there was no difference among these genes involved in soil recalcitrant C degradation (*lignin*). Thus, functional genes involved in labile C degradation probably respond to elevation changes faster than genes involved in recalcitrant C degradation.

### 3.5. The Linkages among Plants, Soil Bacterial Diversity, and Environmental Factors

The relationship among plants, soil bacterial diversity, and environmental factors was analyzed by Pearson’s correlation. Plant and soil bacterial Shannon diversity had no direct positive relationship (*r* = −0.147, *p* = 0.198) (Figure 4a). Soil bacterial Shannon diversity was significantly negatively (*p* < 0.01) related with soil site elevation, MAP, soil moisture, SOC, TN, TP, AN, and NH_4_^+^-N, and significantly positively correlated with MAT and soil pH (*p* < 0.01, Appendix A). However, none of the climatic or soil nutrient environmental parameters were shown to be substantially associated to plant Shannon diversity (Appendix A). Therefore, there was no correlation between plant and soil bacterial alpha diversity, which was probably influenced by diverse environmental conditions.

Partial Mantel tests were performed to determine the relationship between soil bacterial beta diversity and plants, as well as how they were affected by environmental factors. The results showed plant beta diversity was significantly linked with soil bacterial beta diversity (*r* = 0.493, *p* < 0.001), MAT (*r* = 0.453, *p* < 0.001), and MAP (*r* = 0.424, *p* < 0.001), while soil bacterial beta diversity was found to be significantly correlated with MAP (*r* = 0.679, *p* < 0.001) and soil pH (*r* = 0.668, *p* < 0.001) (Table 3). Regression analysis revealed that plant and soil bacterial beta diversity were significantly associated (*r* = 0.659, *p* < 0.001, Figure 4b). CCA results indicated that MAT and MAP were the most important factors determining plant community variations (Figure 5a), whereas MAP and soil pH were the most important variables regulating soil bacterial community variation (Figure 5b). Therefore, MAP and soil pH were the key variables influencing soil bacterial beta diversity. Soil bacterial beta diversity, MAT, and MAP may be the key factors shaping plant beta diversity in the alpine meadow ecosystems.

## 4. Discussion

The primary concerns of whether soil microbial diversity changes along elevation gradients and whether this pattern resembles that observed in macroorganisms remain unanswered. In the current study, plant and soil bacterial diversity had different elevation diversity patterns. Previous studies yielded different findings on microbial community patterns with increasing elevation: a monotonous decrease [23], or increase [44], or no trend [45], or a humpbacked trend [46]. These different elevation distribution patterns of soil microbes might be caused due to the large variety of different environmental conditions [15].

Elucidating the causes of changes in soil microbial diversity and plant diversity, as well as soil microbe–plant interactions under environmental changes remains a research priority that will shed light on significant ecosystem functions such as plant net primary productivity and soil C storage [7,47,48]. Plants’ diversity patterns have been attributed to spatial factors, climate factors, abiotic factors, or biotic factors in previous studies, and these patterns generally exhibit either monotonically declining or hump-shaped diversity patterns with increased elevation [49,50,51]. Plant alpha diversity had no strong relationship with soil bacterial alpha diversity and other environmental factors in the current study, although it may be affected by human disturbances [52]. Soil bacterial Shannon index was closely related with climate factors (elevation, MAT, and MAP) and soil nutrients (soil moisture, pH, and SOC). Previous experimental and observational investigations found some favorable connections between plant and soil microbial alpha diversity [5,8,53,54]. Zhelezova et al. concluded that soil bacterial alpha diversity in the upper horizon was higher in the growing season (vegetation development) and decreased significantly in non-growing season [55]. Yashiro et al. found soil microbial community and plant alliances tended to be synchronized in community turnover across the grassland mountain landscape in the western Swiss Alps [16]. The contradictory results may be explained by discrepancies in the spatial scale of inquiry into environmental variables, as well as resistance or resilience rates of soil microbes to environmental changes [2,41,56].

Shifts in the soil microbial community are important response mechanisms that can help the adaptation of soil ecosystem functions to environmental changes [7,56,57,58]. In this study, SOC and several other soil nutrients decreased dramatically as elevation decreased, resulting in nutrient-rich or nutrient-limited environments at high-elevation and low-elevation sites (SJY-3490 and SJY-3420, respectively). The most important dominating phylum shifted from *Proteobacteria* at higher elevation to *Acidobacteria* at lower elevation (SJY-3490 and SJY-3420), probably in response to their well-known ecological preference for nutrient-rich or nutrient-limited conditions. *Acidobacteria* was known as a good competitor in oligotrophic environments, which could thrive in nutrient-limited ecosystems, while *Proteobacteria* was inclined to be in copiotrophic conditions, which dominated in nutrient-rich environments [1,59,60]. *Acidobacteria* was the most common group at low-elevation, possibly due to their low C turnover, which allows them to adapt to the low-nutrient soil environment as “stress tolerators” [61]. Of course, not all taxa in a phylum will be either copiotrophic or oligotrophic [60], hence other classifier levels should be used to predict soil C loss [62]. In our study, *Beta*- and *Gamma*-*proteobacteria* decreased as elevation decreased; however, *Acidobacteria GP4* and *GP6* increased. Therefore, these classes exhibit obvious adaptability to varying soil nutrient levels.

A significant positive relationship (*p* < 0.01) between soil microbes and plants was encouraged by their beta diversity, suggesting that plant diversity predicted beta but not alpha diversity of soil microbes, confirming Prober et al.’s findings [2]. However, the relationship between plant diversity and soil bacterial diversity is influenced by different environmental drivers. Soil bacterial beta diversity was significantly related with elevation, MAP, and soil pH; nevertheless, plant beta diversity was significantly correlated with soil bacterial beta diversity and climate factors (MAT and MAP). The strong driving role of soil pH, significantly affected by MAP, in soil microbial community composition is consistent with previous work [63,64]. Reduced MAP and soil dryness may improve oxygen availability and soil C cycling [65]. Understanding how soil microbe–plant interactions respond to climate change is a research priority, but findings remain inconsistent [58]. Our results showed soil bacterial beta diversity was one of the key factors influencing plant community structure, implying that a soil microbial-mediated nutrient acquisition strategy could alter local plant diversity directly or indirectly under climate change [66]. Because soil microbes regulate soil nutrient transformations, this may have large implications for plant community composition and ecosystem functions [67]. Soil microbes have been increasingly considered as important drivers of plant diversity in recent years [2,7,9,68]. Van der Heijden et al. predicted that soil microbes promoted plant diversity by increasing soil available nutrients [9]. Delgado-Baquerizo et al. reported that soil microbial diversity played a positive role in linking to multifunctionality in terrestrial ecosystems [69]. Teste et al. identified that the soil microbial nutrient acquisition strategy was an important trait explaining how different responses to soil biota improve local plant diversity [3]. Ferreira DA et al. highlighted a potential overall helper effect of soil biodiversity on plant–arbuscular mycorrhizal fungi symbiosis [68].

Soil microbial communities have important effects on maintaining multiple ecosystem services and functions simultaneously [70,71]. They are sensitive to environmental changes, and alterations in microbial abundance or diversity can have an impact on ecosystem processes and functions [9,69,70]. In our study, the relative abundance of microbial functional genes involved in N and P cycling was significantly higher at the low-elevation sites (SJY-3490 and SJY-3220) than at the SJY-4790 site, suggesting that soil microbial functions related to N and P cycles might have a faster change than C degradation functions. Soil N and P contents are essential limiting factors for plant primary production and microbially-mediated biogeochemistry cycles in terrestrial ecosystems, especially in nutrient-limited environments [72,73,74]. More soil N may be lost from the soils when elevation decreasing or climate warming, mediated by potential denitrification processes, and more N mineralization may occur to meet the needs of ecosystem functions [74,75]. The *amoA* and *hao* genes related to N nitrification were significantly higher in SJY-3490 and SJY-3220 than in SJY-4790. The *nirB*, *nirA*, and *NirR* genes related to assimilatory N reduction, as well as the *nifH* gene related to N_2_ fixation, displayed similar trends along elevation gradient to the *amoA* genes. Upregulation of these microbial functional genes might produce more NH_4_^+^ and NO_3_^−^ to meet the needs of both microorganisms and plants [57,69,74,75]. P availability not only influenced soil microbial growth, but it was also a key factor driving biogeochemical function and microbial community structure [76,77]. The *ppk* gene (polyphosphate kinase) for polyphosphate biosynthesis and the *ppx* gene (exopolyphosphatase) for inorganic polyphosphate degradation were also significantly higher at the SJY-3490 and SJY-3220 sites than at other sites, indicating that more inorganic P might be used by desorption and dissolution from soil minerals to meet ecosystem functions at these sites [69,78].

In summary, we analyzed plant and soil microbial diversity in the alpine meadow ecosystem of the Qinghai–Tibetan Plateau, China, based on Illumina sequencing and GeoChip technologies to better understand the distribution patterns of plant and soil microbial diversity along elevation gradients and their linkages. We realized that plant and soil microbial alpha diversity had different distribution patterns and no links. Functional gene categories involved in N and P cycling may experience quicker elevational gradient shifts than functional genes involved in C degradation. Beta diversity of plants and soil bacteria was significantly increased as elevation distance increased and they were significantly correlated with each other. The elevation, MAP, and soil pH were important environmental variables in shaping soil bacterial beta diversity, while plant beta diversity was significantly correlated with soil bacterial beta diversity and climate factors.

## Figures and Tables

**Figure 1 microorganisms-10-01049-f001:**
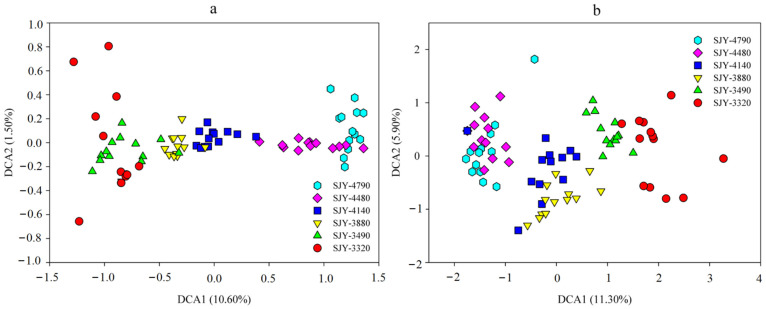
Detrended correspondence analysis of plant (**a**) and soil bacterial (**b**) community structure.

**Figure 2 microorganisms-10-01049-f002:**
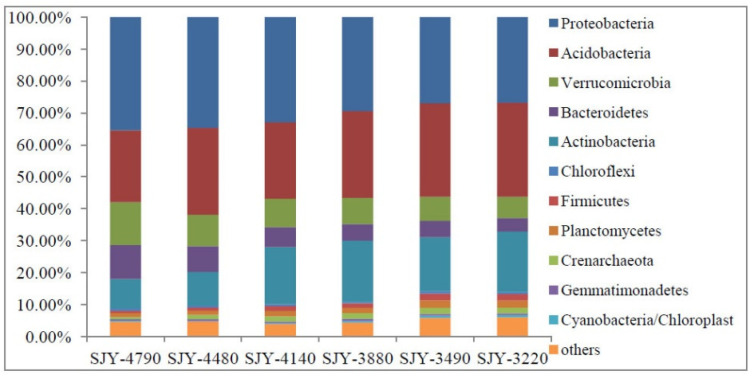
Soil bacterial relative abundance at the phylum level along elevation gradient.

**Figure 3 microorganisms-10-01049-f003:**
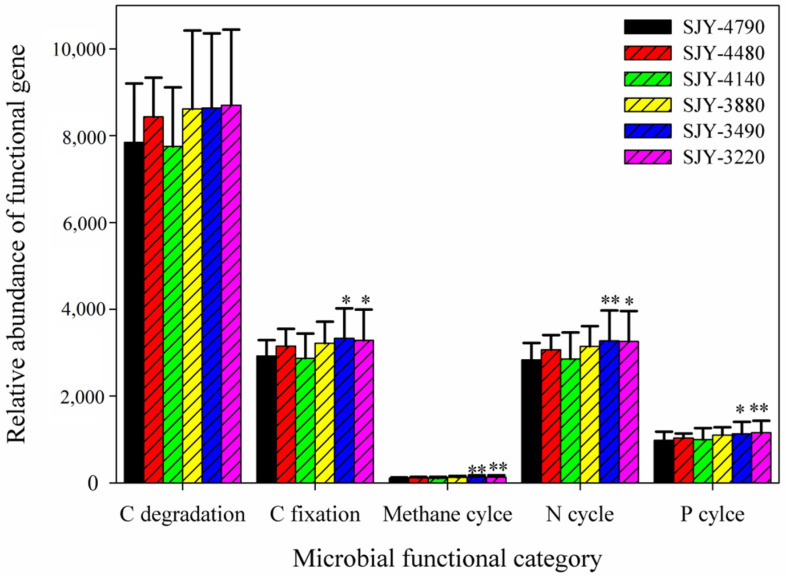
The relative abundance of microbial functional genes involved in carbon (C), nitrogen (N), and phosphorus (P) cycles in all six sampling sites. The relative abundance of each gene category was the sum of detected individual gene signal intensity. All data are presented as mean and standard error. Significant differences between different study sites are indicated above the bars. * *p* <0.1; ** *p* <0.05.

**Figure 4 microorganisms-10-01049-f004:**
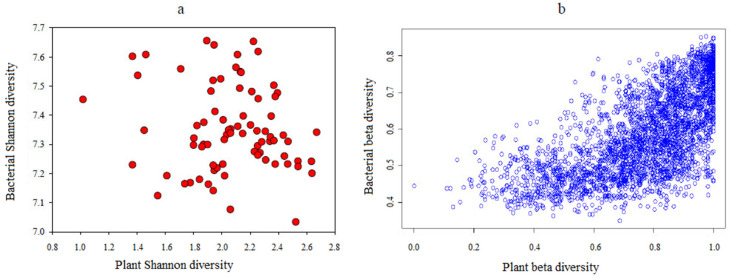
The relationship between alpha (**a**) (r = −0.147, *p* = 0.198) and beta (**b**) (r = 0.659, *p* = 0.001) diversity of plants and bacteria.

**Figure 5 microorganisms-10-01049-f005:**
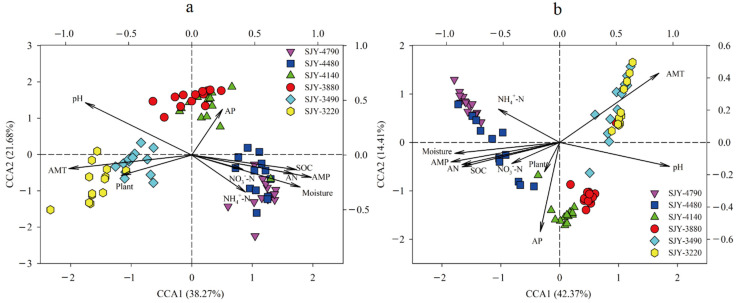
CCA analysis of plants (**a**) and bacteria (**b**) with different environmental factors.

**Table 1 microorganisms-10-01049-t001:** Study sites information along elevation gradient in the Qinghai–Tibet Plateau.

Site Name	Location	Elevation	Latitude, Longitude	Slope	Aspect	Dominant Plant Species	MAT of the Warmest Quarter (°C)	MAP (mm)
SJY-4790	Maduo County, Qinghai Province	4790 m	34°08′16″ N 97°40′22″ E	15°	N	Kobresia tibetica, Kobresia pygmaea	4.90	417.00
SJY-4480	Maduo County, Qinghai Province	4480 m	34°22′15″ N 97°56′57″ E	18°	30°, NW	Kobresia tibetica	6.20	386.00
SJY-4140	Maduo County, Qinghai Province	4140 m	35°24′28″ N 99°21′6″ E	5°	20°, NW	Kobresia pygmaea, Kobresia humilis	6.30	372.00
SJY-3880	Xinghai County, Qinghai Province	3880 m	35°41′26″ N 99°33′1″ E	15°	30°, NW	Kobresia pygmaea, Stipa capillata	7.40	354.00
SJY-3490	Xinghai County, Qinghai Province	3490 m	35°40′10″ N 99°55′13″ E	5°	N	Poa annua, Stipa capillata	10.00	344.00
SJY-3220	Xinghai County, Qinghai Province	3220 m	35°56′6″ N 100°5′27″ E	5°	N	Elymus nutans, Stipa capillata	11.60	319.00

Mean annual precipitation: MAP, mean annual Temperature: MAT.

**Table 2 microorganisms-10-01049-t002:** The plant and soil bacterial alpha diversity in all six study sites.

Site Name	Plant Shannon Index	Plant Pielou Index	Bacterial Shannon Index	Bacterial Simpson Index
SJY-4790	1.97 ± 0.26 c	0.79 ± 0.09 ab	7.27 ± 0.10 a	448 ± 102 a
SJY-4480	2.34 ± 0.23 a	0.84 ± 0.07 a	7.26 ± 0.12 a	483 ± 109 a
SJY-4140	2.27 ± 0.25 ab	0.85 ± 0.08 a	7.35 ± 0.09 a	518 ± 112 a
SJY-3880	1.72 ± 0.30 d	0.74 ± 0.10 b	7.27 ± 0.12 a	476 ± 87 a
SJY-3490	2.09 ± 0.14b	0.85 ± 0.04a	7.50 ± 0.08b	755 ± 145b
SJY-3220	2.05 ± 0.34 bc	0.86 ± 0.08 a	7.45 ± 0.16 b	702 ± 185 b

Data present the mean value and standard error. Significant differences among study sites are indicated by alphabetic letters. *p* < 0.05.

**Table 3 microorganisms-10-01049-t003:** Relationship between plant or soil bacterial beta diversity and climate or soil environmental factors estimated using partial Mantel tests.

Environmental Factors	Plant Beta Diversity		Bacterial Beta Diversity	
	r	*p*	r	*p*
Site elevation	0.496	<0.001	0.670	<0.001
Mean annual temperature	0.410	0.001	0.437	0.001
Mean annual precipitation	0.373	0.001	0.624	0.001
Soil moisture	0.164	0.001	0.452	0.001
Soil pH	0.255	0.001	0.634	0.001
Soil organic carbon	0.036	0.107	0.165	0.001
Available nitrogen	−0.017	0.735	0.094	0.008
Available phosphorus	0.089	0.010	0.030	0.268
Soil NH^4+^- N	−0.103	0.999	0.150	0.003
Soil NO^3−^- N	−0.039	0.864	−0.040	0.778
Bacteria Shannon index	0.176	0.004	-	-
Plant Shannon index	-	-	0.060	0.069
Bacterial beta diversity	0.493	0.001	-	
Plant beta diversity	-	-	0.493	0.001

## Data Availability

16S rRNA sequences have been deposited to GenBank databases and the accession number is SRP117325.

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
