# Peer review of "Distinct Elevational Patterns and Their Linkages of Soil Bacteria and Plant Community in An Alpine Meadow of the Qinghai–Tibetan Plateau"

_microorganisms, 2022, doi:10.3390/microorganisms10051049_

Round 1
Reviewer 1 Report
Microorganisms
Title: Distinct elevational patterns and their linkages of soil bacteria and plant community in an alpine meadow ecosystem
Author: Jing Cong
General Comments
The paper examines plant community composition, soil microbial community composition, and a few microbial genes along an elevation gradient in China. This is not a new idea. Nevertheless, the data are interesting.
The paper is generally readable. I have only a few minor comments and suggestions.
Specific Comments
- The Introduction reads well. However, consider adding one more paragraph in which you state specifically how you will show that soil microorganisms 'drive' plant community composition. You make the claim but never say how this is done. Typically, it requires an experimental manipulation, and showing in a gradient study is not obvious.
- Indeed, you might change the tone of the Introduction. Rather that argue plants drive soil microorganisms or vice versus, just say that plants and soil microorganisms develop relationships with each other. These relationships tend to be mutualistic. Therefore, you can ask whether the mutualistic relationships persist along the elevational gradient. It would be interesting to know if relationships persist with climate change.
- The Methods read well.
- The Results are presented concisely.
- The Discussion is okay. My one concern is some strong wording, such as plant diversity 'forecasts' beta and alpha diversity of soil microorganisms. I agree that plants and microorganisms have different diversity patterns, but I do not see how you concluded one drives the other. This requires an experiment.
Technical Comments
- Title: consider deleting 'ecosystem.' The study considers community ecology more so than ecosystem ecology.
- Line 35: rather than 'drive' perhaps say 'can influence.'
- Line 36: delete 'influence.'
- Line 43: rather than say 'restricted' consider saying 'that we still have much to learn.'
- Line 55: the wording is awkward. Indeed, it is not clear what you are trying to say. Revise.
- Line 58: change the wording to say, that 'elevational gradients can serve as analogs for climate change.'
- Line 100: indicate these are described below or give the methods here.
- Table 2. Round values for Bacterial Simpson Index to the nearest integer. You cannot measure accurately to a fraction of one.
- Line 299: consider changing 'no significant' to 'different.'
Author Response
General Comments
The paper examines plant community composition, soil microbial community composition, and a few microbial genes along an elevation gradient in China. This is not a new idea. Nevertheless, the data are interesting.
The paper is generally readable. I have only a few minor comments and suggestions.
Response: Thank you for your letter and for the reviewers’ comments concerning our manuscript. Those comments are all valuable and very helpful for revising and improving our paper, as well as the important guiding significance to our researches. We have studied comments carefully and have made correction which we hope meet with approval. Revised portion are marked in red in the paper. The main corrections in the paper and the responds to the reviewer’s comments are as following.
Specific Comments
- The Introduction reads well. However, consider adding one more paragraph in which you state specifically how you will show that soil microorganisms 'drive' plant community composition. You make the claim but never say how this is done. Typically, it requires an experimental manipulation, and showing in a gradient study is not obvious.
Indeed, you might change the tone of the Introduction. Rather that argue plants drive soil microorganisms or vice versus, just say that plants and soil microorganisms develop relationships with each other. These relationships tend to be mutualistic. Therefore, you can ask whether the mutualistic relationships persist along the elevational gradient. It would be interesting to know if relationships persist with climate change.
Response: Special thanks to you for your good comments. We have changed the tone of the introduction as “Evidence is increasing suggesting soil microbes can influence plant diversity and community composition, because soil microbes are important in biogeochemical cycles and many ecosystem processes [6-8]. Free-living soil microbes might indirectly promote plant diversity by increasing the diversity of available nutrient pools [9]. Both arbuscular mycorrhizal fungi and N-fixing rhizobia bacteria symbionts complement each other, resulting in increased plant diversity [10]. Therefore, the relationships between soil microbial diversity and plant diversity tend to mutualistic. However, there is still inconsistent, especially for somewhat disengaged from alpha to beta diversity [1,2,11,12]. As a result, our knowledge of the underlying links between soil microbial diversity and plant diversity remains that we still have much to learn [2]”. The revised part is marked in red in revised manuscript.
- The Methods read well.
Response: Special thanks to you for your good comments.
- The Results are presented concisely.
Response: Special thanks to you for your good comments.
- The Discussion is okay. My one concern is some strong wording, such as plant diversity 'forecasts' beta and alpha diversity of soil microorganisms. I agree that plants and microorganisms have different diversity patterns, but I do not see how you concluded one drives the other. This requires an experiment.
Response: Special thanks to you for your good comments. We have revised the similar strong words in our manuscript including changing ‘forecasts’ to ‘predict’.
- Technical Comments
5.1 Title: consider deleting 'ecosystem.' The study considers community ecology more so than ecosystem ecology.
Response: Thank you for your suggestion. We have revised it as“Distinct elevational patterns and their linkages of soil bacteria and plant community in an alpine meadow of the Qinghai-Tibetan Plateau”.
5.2 Line 35: rather than 'drive' perhaps say 'can influence.'
Response: Thank you for your suggestion. We have revised it.
5.3 Line 36: delete 'influence.'
Response: Thank you for your suggestion. We have deleted it.
5.4 Line 43: rather than say 'restricted' consider saying 'that we still have much to learn.'
Response: Thank you for your suggestion. We have revised it.
5.5 Line 55: the wording is awkward. Indeed, it is not clear what you are trying to say. Revise.
Response: Thank you for your suggestion. We have revised it as ‘numerous warming studies focus only on several temperature points without continuous gradient changes’.
5.6 Line 58: change the wording to say, that 'elevational gradients can serve as analogs for climate change.'
Response: Thank you for your suggestion. We have revised it as ‘Elevation gradients can serve as analogs for climate change, characterized by biotic turnover over small geographic distances and dramatic changes in climate. They play a critical role in the progression of biogeographical and ecological studies, as well as the prediction of possible consequences of long-term climate change’.
5.7 Line 100: indicate these are described below or give the methods here.
Response: Thank you for your suggestion. We have revised it as “These detailed analyses are described in the following”.
5.8 Table 2. Round values for Bacterial Simpson Index to the nearest integer. You cannot measure accurately to a fraction of one.
Response: Thank you for your suggestion. We have revised the Table 2 and the text in revised manuscript.
5.9 Line 299: consider changing 'no significant' to 'different.'
Response: Thank you for your suggestion. We have revised it, changing 'no significant' to 'different.'.
Reviewer 2 Report
The present article is devoted to the topical problem of studying soil bacteria along height gradient in the ecosystem of alpine meadows. This work is needed to shed light on important functions of mountain ecosystems. The results of authors quite fully and in detail explain why there were no relationships between the alpha diversity of bacteria and plants of the alpine meadows of the Qinghai-Tibet Plateau. In addition, the study showed why soil pH and average annual rainfall were important environmental variables influencing soil bacterial diversity in alpine grasslands of the Qinghai-Tibet Plateau. The studies were performed using a number of modern molecular biological methods and excellent statistical processing, so the results obtained are beyond doubt. At the same time, the article has some shortcomings:
1 - The title of the article is too general. It seems that the authors wanted to generalize their results to all alpine meadow ecosystems. However, the authors analyzed only the ecosystems of alpine meadows in the Qinghai-Tibet Plateau. Therefore, the authors should reflect the geographical reference of the place of study in the title of the article.
2 - The number of keywords should be increased because they do not yet contain information about research methods.
3 - The authors did not take into account the profile structure of the soil along the horizons. In further studies, the authors should take samples not by depth, but by soil horizons, since they usually have different properties.
4 - It is not entirely clear why the authors classify Proteobacteria as copiotrophs and Acidobacteria as oligotrophs. These types of bacteria are highly heterogeneous and contain many ecological groups. Thus, this statement of the authors is not true. Therefore, I advise the authors to come up with another explanation for the discussion regarding the reason for the decrease in the number of Acidobacteria with decreasing altitude.
5 - It seems strange that the authors did not find a correlation between plant and soil bacterial alpha diversity. There are many studies that show a reliable and direct dependence of the vegetation cover on the soil microbiome. For example, authors should read and cite Zhelezova, A.; Chernov, T.; Nikitin, D.; Tkhakakhova, A.; Ksenofontova, N.; Zverev, A.; Kutovaya, O.; Semenov, M. Seasonal Dynamics of Soil Bacterial Community under Long-Term Abandoned Cropland in Boreal Climate. Agronomy 2022, 12, 519. https://doi.org/10.3390/agronomy12020519
6 - The authors do not indicate whether they took samples in rhizosphere or non-rhizosphere soil loci. However, it is known that the composition of the rhizosphere microbiome is extremely different from the structure of the native soil microbiome. Therefore, the authors should read and cite the article by Semenov et al.: Semenov, M. V., Nikitin, D. A., Stepanov, A. L., & Semenov, V. M. (2019). The structure of bacterial and fungal communities in the rhizosphere and root-free loci of gray forest soil. Eurasian Soil Science, 52(3), 319-332. https://doi.org/10.1134/S1064229319010137
6 - The authors give reasoning about the ecological and indicator functions of the soil and the role of microorganisms in it. I believe, therefore, the authors should read and cite the work of Nikitin and co-authors: Nikitin, D. A., Semenov, M. V., Chernov, T. I., Ksenofontova, N. A., Zhelezova, A. D., Ivanova, E. A., Khitrov, N. B. & Stepanov, A. L. (2022). Microbiological Indicators of Soil Ecological Functions: A Review. Eurasian Soil Science, 55(2), 221-234. https://doi.org/10.1134/S1064229322020090
Author Response
Comments and Suggestions for Authors
The present article is devoted to the topical problem of studying soil bacteria along height gradient in the ecosystem of alpine meadows. This work is needed to shed light on important functions of mountain ecosystems. The results of authors quite fully and in detail explain why there were no relationships between the alpha diversity of bacteria and plants of the alpine meadows of the Qinghai-Tibet Plateau. In addition, the study showed why soil pH and average annual rainfall were important environmental variables influencing soil bacterial diversity in alpine grasslands of the Qinghai-Tibet Plateau. The studies were performed using a number of modern molecular biological methods and excellent statistical processing, so the results obtained are beyond doubt. At the same time, the article has some shortcomings:
Response: Thank you for your letter and for the reviewers’ comments concerning our manuscript. Those comments are all valuable and very helpful for revising and improving our paper, as well as the important guiding significance to our researches. We have studied comments carefully and have made correction which we hope meet with approval. Revised portion are marked in red in the paper. The main corrections in the paper and the responds to the reviewer’s comments are as following.
1 - The title of the article is too general. It seems that the authors wanted to generalize their results to all alpine meadow ecosystems. However, the authors analyzed only the ecosystems of alpine meadows in the Qinghai-Tibet Plateau. Therefore, the authors should reflect the geographical reference of the place of study in the title of the article.
Response: Thank you for your suggestion. We have made correction according to the Reviewer’s comments. The title was revised by ‘Distinct elevational patterns and their linkages of soil bacteria and plant community in an alpine meadow of the Qinghai-Tibetan Plateau’.
2 - The number of keywords should be increased because they do not yet contain information about research methods.
Response: Thank you for your suggestion. We have made correction according to the Reviewer’s comments. We have added the keywords ‘GeoChip 4.0, 16S rRNA sequencing;’ about research methods.
3 - The authors did not take into account the profile structure of the soil along the horizons. In further studies, the authors should take samples not by depth, but by soil horizons, since they usually have different properties.
Response: Thank you for your suggestion. Previous samples were mainly collected from non-rhizosphere soil in 0-10cm soil depth from six study sites. These samples were taken and mixed with a horizontal expansion at 5-, 10-, 20-, 50-, 100-, and 200-m distances from the original point in grid of 200 m × 200 m. Actually, we only mixed with these different horizontal soils as a whole to further explore, however, we did not take these single horizontal soils into account the soil microbial analysis. In the early time, we probably considered that there were relatively few types of aboveground vegetation with little variation over short distances. Therefore, we did not compare the profile structure of the soil along the horizons. In the future work, we will pay more attention on taking samples by soil horizons. Special thanks to you for your good suggestions.
4 - It is not entirely clear why the authors classify Proteobacteria as copiotrophs and Acidobacteria as oligotrophs. These types of bacteria are highly heterogeneous and contain many ecological groups. Thus, this statement of the authors is not true. Therefore, I advise the authors to come up with another explanation for the discussion regarding the reason for the decrease in the number of Acidobacteria with decreasing altitude.
Response: Thank you for your suggestion. We used the words ‘oligotrophs’ and ‘copiotrophs’ by referring the early study (Kuznetsov SI et al, 1979). It defined that an oligotrophic microflora predominating under natural conditions possessed a high rate of growth at low concentrations of organic substances (Kuznetsov SI et al, 1979), and vice versa for copiotrophs. Another paper ‘The Ecology of Acidobacteria: Moving beyond Genes and Genomes’ depicted that the strong negative correlation between the abundance of Acidobacteria and concentration of organic carbon in soil has led to the conclusion that members of this phylum may be oligotrophic bacteria (Anna MK et al, 2016), and other paper also reported the conclusion(Ryan TJ et al, 2009; Akane C et al, 2021). However, it should point out that not necessarily all members would be oligotrophic (Fierer et al, 2007). Therefore, our description was not accurate enough. We have revised it as ‘The most important dominating phylum shifted from Proteobacteria at higher elevation to Acidobacteria at lower elevation (SJY-3490 and SJY-3420), probably in response to their well-known ecological preference for nutrient-rich or nutrient-limited conditions. Acidobacteria was known as a good competitor in oligotrophic environments, which could thrive in nutrient-limited ecosystems, while Proteobacteria was inclined to be in copiotrophic conditions, which dominated in nutrient-rich environments [1,59,60]. Acidobacteria was the most common group at low elevation, possibly due to their low C turnover, which allows them to adapt to the low-nutrient soil environment as “stress tolerators” [61]. Of course, not all taxa in a phylum will be either copiotrophic or oligotrophic [60], hence other classifier levels should be used to predict soil C loss [62]’, in revised manuscript.
5 - It seems strange that the authors did not find a correlation between plant and soil bacterial alpha diversity. There are many studies that show a reliable and direct dependence of the vegetation cover on the soil microbiome. For example, authors should read and cite Zhelezova, A.; Chernov, T.; Nikitin, D.; Tkhakakhova, A.; Ksenofontova, N.; Zverev, A.; Kutovaya, O.; Semenov, M. Seasonal Dynamics of Soil Bacterial Community under Long-Term Abandoned Cropland in Boreal Climate. Agronomy 2022, 12, 519. https://doi.org/10.3390/agronomy12020519
Response: Thank you for your suggestion. We have read and cited the research as “Zhelezova et al. concluded that soil bacterial alpha-diversity in the upper horizon was higher in growing season (vegetation development), and decreased significantly in non-growing season [55].” in revised manuscript.
6 - The authors do not indicate whether they took samples in rhizosphere or non-rhizosphere soil loci. However, it is known that the composition of the rhizosphere microbiome is extremely different from the structure of the native soil microbiome. Therefore, the authors should read and cite the article by Semenov et al.: Semenov, M. V., Nikitin, D. A., Stepanov, A. L., & Semenov, V. M. (2019). The structure of bacterial and fungal communities in the rhizosphere and root-free loci of gray forest soil. Eurasian Soil Science, 52(3), 319-332. https://doi.org/10.1134/S1064229319010137
Response: Thank you for your suggestion. We took samples in non-rhizosphere soil. We have read and cite the article as “Soil samples were obtained at 0-10 cm soil depth in non-rhizosphere soil of each plot using the diagonal method [37]” in our revised manuscript.
7 - The authors give reasoning about the ecological and indicator functions of the soil and the role of microorganisms in it. I believe, therefore, the authors should read and cite the work of Nikitin and co-authors: Nikitin, D. A., Semenov, M. V., Chernov, T. I., Ksenofontova, N. A., Zhelezova, A. D., Ivanova, E. A., Khitrov, N. B. & Stepanov, A. L. (2022). Microbiological Indicators of Soil Ecological Functions: A Review. Eurasian Soil Science, 55(2), 221-234. https://doi.org/10.1134/S1064229322020090
Response: Thank you for your suggestion. We have read and cited the work as ‘soil microbial communities have important effects on maintaining multiple ecosystem services and functions simultaneously [70,71].’ in our revised manuscript.